# In Vitro Analysis of Superparamagnetic Iron Oxide Nanoparticles Coated with APTES as Possible Radiosensitizers for HNSCC Cells

**DOI:** 10.3390/nano13020330

**Published:** 2023-01-12

**Authors:** Clara Emer, Laura S. Hildebrand, Bernhard Friedrich, Rainer Tietze, Rainer Fietkau, Luitpold V. Distel

**Affiliations:** 1Department of Radiation Oncology, University Hospital Erlangen, Friedrich-Alexander-Universität Erlangen-Nürnberg, 91054 Erlangen, Germany; 2Comprehensive Cancer Center Erlangen-EMN (CCC ER-EMN), 91054 Erlangen, Germany; 3ENT-Department, Else Kröner-Fresenius-Stiftung Professorship, Section for Experimental Oncology and Nanomedicine (SEON), University Hospital Erlangen, 91054 Erlangen, Germany

**Keywords:** nanoparticles, SPION, HNSCC, head and neck cancer cell lines, cell survival, doubling time, time lapse

## Abstract

Superparamagnetic iron oxide nanoparticles (SPION) are being investigated for many purposes, e.g., for the amplification of ionizing radiation and for the targeted application of therapeutics. Therefore, we investigated SPIONs coated with (3-Aminopropyle)-Triethoxysilane (SPION-APTES) for their influence on different head and neck squamous cell carcinoma (HNSCC) cell lines, as well as for their suitability as a radiosensitizer. We used 24-well microscopy and immunofluorescence microscopy for cell observation, growth curves to determine cytostatic effects, and colony formation assays to determine cytotoxicity. We found that the APTES-SPIONs were very well taken up by the HNSCC cells. They generally have a low cytotoxic effect, showing no significant difference in clonogenic survival between the control group and cells treated with 20 µg Fe/mL (*p* > 0.25) for all cell lines. They have a cytostatic effect on some cell lines cells (e.g., Cal33) that is visible across different radiation doses (1, 2, 8 Gy, *p* = 0.05). In Cal33, e.g., SPION-APTES raised the doubling time at 2 Gy from 24.53 h to 41.64 h. Importantly, these findings vary notably between the cell lines. However, they do not significantly alter the radiation effect: only one out of eight cell lines treated with SPION-APTES showed a significantly reduced clonogenic survival after ionizing radiation with 2 Gy, and only two showed significantly reduced doubling times. Thus, although the APTES-SPIONs do not qualify as a radiosensitizer, we were still able to vividly demonstrate and analyze the effect that the APTES-SPIONs have on various cell lines as a contribution to further functionalization.

## 1. Introduction

Superparamagnetic iron oxide nanoparticles (SPIONs) have proven beneficial in the medical field: they are controllable by a magnetic field and can be actively steered in the body. SPIONs are being investigated and used for a plethora of different applications, including the delivery of siRNA in gene therapy, the targeted delivery of chemotherapeutics, induction of apoptosis in cancer cells through magnetic hyperthermia, and as an MRI contrast agent [1,2,3,4,5]. SPIONs coated with (3-Aminopropyle)-Triethoxysilane (SPION-APTES), due to the free amino groups on their surface, offer the possibility to link various therapeutic agents and substances for different medical applications [6,7,8]. These characteristics make them a prime candidate for many applications but particularly for new approaches in tumor therapy, where they have already been studied for the targeted application of chemotherapeutics, growth-inhibitors, or antibodies on cancers such as cervical carcinoma, lymphoma, and ovarian cancer [9,10,11]. They have even proven suitable to deliver agents for photodynamic therapy [12,13]. These broad application possibilities and advantages of SPIONs with an APTES coating were the reason for us to conduct our study with SPION-APTES. SPION-APTES have, however, not been studied for their usability as radiosensitizers, even though other kinds of SPIONs have already proven to increase the effect of ionizing radiation [14,15,16,17,18,19].

A prerequisite for the use of any SPION is knowledge of their effect on cells in order to protect them from unwanted cytotoxicity [20]. However, the characteristics and effects of SPIONs on cells vary greatly, depending on their size, magnetic property, and coating; so, they must be determined for each kind of SPION [3].

In head and neck squamous cell carcinoma (HNSCC), radiation is an important part of therapy because surgery is often not an option due to the difficult anatomical conditions of the area [21,22]. However, radiation therapy is also limited by the proximity to critical organs, which must be protected from high radiation doses to minimize morbidity [23,24]. SPIONs could help to increase the therapeutic effect of ionizing radiation as a targeted radiosensitizer at the same radiation dose without additional impact on the surrounding structures and without the side effects of the existing radiosensitizers such as cisplatin [25,26].

As early as 2010, Hainfeld et al. showed in a murine model that gold nanoparticles enhance the effect of ionizing radiation on HNSCC, which was later confirmed by further studies for gold nanoparticles, as well as for gold-coated SPIONs [27,28,29]. Last year, two more publications showed the potential of nanoparticles as radiosensitizers for HNSCC. However, it should be noted that these were not iron oxide nanoparticles and that they were not pure nanoparticles but nanoparticle formulations, so that, in contrast to our study, the radiation-enhancing property cannot be solely attributed to the nanoparticles [30,31].

In our study, we investigated the interaction of SPION-APTES with HNSCC cells and the cytotoxic and cytostatic effects that the SPION-APTES have on the cells. We were particularly interested in whether they enhance the effect of ionizing radiation and can therefore be used as a radiosensitizer.

## 2. Materials and Methods

### 2.1. Cell Culture and SPIONs

We used eight cell lines in total: five HPV-negative (HPV-) HNSCC cell lines (Cal33, CLS 354, Detroit 562 (Det 562), HSC-4, and RPMI 2650) and two HPV-positive (HPV+) HNSCC cell lines (UD-SCC-2 and UM-SCC-47) containing HPV-16 (one of the most common high-risk HPV types) and one cell line of healthy bronchial epithelium (BEAS-2B). Cal33, HSC-4, UD-SCC-2, and UM-SCC-47 were provided by Dr. Thorsten Rieckmann (University Medical Center Hamburg-Eppendorf, Germany); CLS 354, Det 562, and RPMI 2650 were purchased from CLS Cell Lines Service GmbH (Eppelheim, Germany); BEAS-2B were provided by the United Kingdom Sigma/Public Health Consortium.

All cells were cultured in Dulbecco’s modified Eagle’s Medium (DMEM; PAN-Biotech GmbH, Aidenbach, Germany) with 10% Fetal bovine serum (FBS; Sigma Aldrich, St. Louis, MI, USA) and 1% penicillin/streptomycin (Thermo Fisher Scientific; Waltham, MA, USA). The cells were grown in a humidified atmosphere at 37 °C and 5% CO_2_.

This study uses SPIONs (Figure 1a) that are coated with APTES, a surface coating that provides vast possibilities for attachment of different molecules through free amino groups (Figure 1b,c). The SPIONs were produced using a one-step coprecipitation process [8,32,33]. Briefly, Iron(II) chloride and iron(III) chloride were dissolved in water in a molar ratio of 1:2. The mixture was stirred while a protective argon atmosphere was applied. The precipitation of iron oxide was induced by the addition of 25% ammonia solution, followed by a temperature increase to 70 °C for 30 min. Thereafter, 3 mL of APTES was added to the mixture, and it was stirred for additional 3 h. After cooling down, the dispersion was stirred three times magnetically with distilled water to remove excess APTES and unreacted products. Particles were redispersed in water and stored at 4 °C.

Iron content of SPION-APTES as well as hydrodynamic size (Z-average) and zeta potential at pH 7.5 was determined following the protocols of Mühlberger et al. [34]. The hydrodynamic size was determined as 166 nm, the zeta potential as +49 mV, and the magnetic susceptibility as 4.1 × 10^3^. As shown in previous publications, the particles proved to be stable at the selected parameters [8,32]. For 3D imaging, SPION-APTES were used, which were functionalized with FITC-BSA (Sigma-Aldrich Chemie GmbH, St. Louis, MI, USA) that was bound to the particles using a well-established functionalization procedure, as presented recently [8]. All SPIONs were stored in distilled H_2_O at 4 °C until use. To obtain the required concentrations, they were diluted with medium.

### 2.2. Monitoring of Cells with 24-Well Microscopy

To evaluate how the cells interact with SPIONs at different concentrations, we monitored them under a 24-well microscope (zenCELL owl; innoME GmbH, Espelkamp, Germany). Cells were seeded in a 24-well plate with 30,000 cells and 0.8 mL of medium per well. Immediately afterwards, monitoring began with a 24-well microscope in the incubator, which acquired one image every hour or quarter of an hour. The cells were left to settle for 24 h, and then SPIONs were added with target concentrations of 1, 5, 10, 20, and 50 µg Fe/mL. To ensure an even distribution, the SPIONs were diluted in 200 µL medium before adding. It should be considered that particles sediment over time. This fact causes a higher concentration on the cells. The number of cells as well as the volume of the medium was always kept constant, so the same concentration always occurred at the cells. The monitoring continued until most of the wells showed confluence of cells, which was after approximately 4 days. After observing the cells at different concentrations of SPION-APTES, we visually determined 20 µg Fe/mL as the optimal concentration for the subsequent experiments, as this was a concentration that did not seem to impact cell growth but resulted in the majority of the cells containing SPION-APTES.

### 2.3. Growth Curves via 24-Well Microscopy

For growth curves with 24-well microscopy, the cells were harvested 48 h prior to the experiment and seeded in tissue flasks. After 24 h, SPIONs were added to one flask at a concentration of 20 µg Fe/mL, and one flask was left without SPIONs as a control group. After 24 h, all cells were harvested and seeded into a 24-well plate with 10,000 cells and 1 mL of medium per well. The cells were irradiated after another 5 h. Dependent on the experiment, they were either irradiated with 2 Gy, with one half of the plate being shielded with a lead plate as a control group, or with doses between 0 and 8 Gy. We used the doses of 0–8 Gy to obtain a broad overview or 2 Gy as a clinically relevant dose [35].

The plate was then placed in the 24-well microscope inside the incubator for monitoring (Figure 2a). The microscope acquired hourly images for 5 days while also automatically capturing the cell number. To evaluate the doubling time, we manually determined the interval during which the cell number increased exponentially; we then fitted an exponential function to the data in this interval:(1)$$y=y0\;\cdot \exp\left(k\;\cdot \;t\right)$$
where *y*0 are the cell numbers at time zero, *k* is the rate constant, and *t* is the time in hours. Doubling time is computed as ln(2)/*k*.

For the graphs, the data were smoothened with a moving average over five data points. Experiments were performed with three replicates.

### 2.4. Cell Imaging by Immunofluorescence Microscopy

For cell imaging by immunofluorescence microscopy, the cells were seeded in rubber chambers on slides and left to settle for 8 h. SPIONs were added to the chambers with 20 µg Fe/mL at different timepoints, so they incubated on the cells for 1, 6, 12, 24, 48, and 72 h. The cells were then fixed with 4% formaldehyde solution (Sigma Aldrich, St. Louis, MI, USA) and blocked with BSA (10 mg/mL in 10% FBS; SERVA Electrophoresis GmbH, GmbH, Germany) overnight. They were stained with a primary anti-α-Tubulin antibody (ab52866; Abcam, Cambridge, UK) diluted at 1:250 with 1% BSA overnight at 4 °C. The next day, Alexa Fluor 555 donkey anti-rabbit (A-31572; Thermo Fisher Scientific, USA) was used as a secondary antibody, diluted at 1:200 with 1% BSA with an incubation period of 2 h at 20 °C in a humidified atmosphere. The images were captured with a fluorescence microscope (Axioplan 2; Zeiss, Göttingen, Germany), and the SPIONs were captured with transmitted light. For 3D imaging, we used SPIONs loaded with FITC fluorochromes and the same staining protocol with the same primary antibody and Alexa Flour 750 donkey anti-rabbit (ab175728; Abcam; Cambridge, UK) as secondary antibody. For the images, we recorded a layered image with the Apotome 3 (Carl Zeiss AG, Oberkochen, Germany) and assembled it with the associated computer software.

### 2.5. Colony Formation Assay

We used the colony formation assay to determine clonogenic cell survival after ionizing radiation, as it is the corresponding gold-standard method [36]. Cells were harvested 48 h prior to the experiment and seeded into tissue flasks, one each for the nanoparticle and control groups. After 24 h, SPIONs were added at a concentration of 20 µg Fe/mL. Twenty-four hours later, the cells were seeded into Petri dishes and irradiated after another 3 h. We chose irradiation doses of either 0–8 Gy to obtain a broad overview or 2 Gy as a clinically relevant dose [35,37]. The cultures were incubated for 2 weeks, and colonies were counted if they contained more than 50 cells (Figure 2b) [38]. The colonies were stained with methylene blue (#66725, Sigma Aldrich, Munich, Germany) for 30 min at room temperature and then counted semi-automatically with image analyzing software (Biomas, MSAB, Erlangen, Germany). Plating efficacy and surviving fraction were calculated. All experiments were performed at least three times independently.

### 2.6. Statistics

Graphs were generated using Microsoft Excel 2016 (Microsoft Corporation, Redmond, DC, USA) and GraphPad prism 8 (GraphPad Software, San Diego, CA, USA). Significance was determined using the unpaired one-tailed Mann–Whitney U-test at a significance level of 0.05.

## 3. Results

We studied the effect of SPION-APTES (Figure 1a) on eight cell lines focusing on their growth behavior. The SPION-APTES have a hydrodynamic diameter of 166 nm with a narrow size distribution, which describes the size of particles dissolved in liquid. They have a zeta potential of +49 mV [8,26]. We also examined whether the SPIONs alter the effect that ionizing radiation has on the cells. Of the eight cell lines, five are HPV- tumor cell lines (Cal33, CLS 354, Det 562, HSC-4, and RPMI 2650), two are HPV+ tumor cell lines (UD-SCC-2 and UM-SCC-47), and one is a healthy bronchial cell line (BEAS-2B).

### 3.1. Reaction of the Cells to SPION-APTES

First, we examined the reaction of the cells to the SPIONs. Observing the cells with a 24-well camera, we saw that their growth was barely affected at concentrations of up to 50 µg Fe/mL compared to a control without SPIONs (video in Appendix A). They also appear to absorb the SPIONs efficiently, resulting in cells appearing black and there being a nanoparticle-free rim around the cells (Figure 3a). With the help of immunofluorescence microscopy, we were able to determine that the SPIONs were inside the cells and not just on the cell surface. In Figure 3b, the SPIONs near the nucleus (lower half of the cell) are seen between the cytoskeleton, without reducing the fluorescence of α-Tubulin (white arrow in fourth panel), while SPIONs closer to the cell margins (upper half of the cell) block the fluorescence of α-Tubulin by lying on the cell (red arrow in fourth panel, α-Tubulin in third panel discontinues and is not visible at the orange marked cell margin). Figure 3c,d also shows SPIONs closer to the nucleus located between the cytoskeleton, while those further away are outside the cell.

### 3.2. Behavior over Time of Cells with SPION-APTES

In the 24-well microscope, we saw that even though the nanoparticles were initially evenly distributed among the cells, clusters of cells with high nanoparticle density formed after some time. The time course of nanoparticle distribution was studied by immunofluorescence microscopy (Figure 4a). We found that the SPIONs, which were still evenly distributed after one hour, mostly adhered to the cells after six hours. After 12 h, there were hardly any free SPIONs left, although most of them were sticking to the outer edges of the cells at this point. After 24 h, almost all cells contained SPIONs, which seemed to be mainly inside the cells. In the following 48 h (cf. images denoted with “48 h” and “72 h” in Figure 4a), we saw the same tendencies we had already observed with the 24-well microscope: some cells accumulated SPIONs, while others hardly contained any. In a 24-well-microscopy experiment with higher recording frequency, we observed that SPIONs previously taken up by a cell were unevenly distributed among the daughter cells after cell division (Figure 4b).

### 3.3. Cytotoxic Effect of SPION-APTES

We then investigated whether the SPIONs themselves have a cytotoxic effect and whether they alter the effect of ionizing radiation on the cells (Figure 5). The colony formation assays (Figure 2a and Figure 5a) performed for this purpose did not show a significant effect for any radiation dose between RPMI 2650 cells that had been treated with SPIONs and those that had not (*p* = 0.050; Figure 5b). The Cal33 cells did show a slight difference between the groups at higher radiation doses, which only became significant at 8 Gy (*p* = 0.050; Figure 5c). We performed colony formation assays with five other cell lines at 0 and 2 Gy. Only the survival fraction of the HSC-4 cells decreased clearly (*p* = 0.050) between the combination of radiation and SPIONs and the cells treated with SPIONs only (Figure 5d–h).

### 3.4. Cytostatic Effect of SPION-APTES

After noting the low cytotoxicity, we examined whether the SPIONs had a cytostatic effect by measuring growth curves (Figure 2b and Figure 6). The growth rates of Cal33 and RPMI 2650 cells differed clearly. There was no difference for the RPMI 2650 cells between the groups treated with SPIONs and those without SPIONs at any radiation dose (*p* ≥ 0.343) (Figure 6a–c). In contrast, Cal33 cells were more sensitive to the SPIONs themselves and especially to the combination of radiation and SPIONs. Thus, after 105 h of nanoparticle incubation, significantly fewer cells were present (*p* = 0.050), and doubling times were significantly higher in several cases (*p* = 0.050) (Figure 6d–f). We also performed growth curves at 0 and 2 Gy for another six cell lines: overall, four out of six lines had a significant increase in doubling times if SPIONs were present. However, this difference was mostly reversed when both the nanoparticle group and control group were irradiated with 2 Gy, with only the BEAS-2B cell line having a clearly increased doubling time (Figure 6g–i).

## 4. Discussion

In this study, we evaluated the cytostatic and cytotoxic effect of SPION-APTES on HNSCC cells and examined their possible use as radiosensitizers.

Our observation of efficient uptake in HNSCC cells is consistent with the results found in the study of Balk et al. with similar SPIONs [39]. The impression that the cells attract the SPIONs seems to be correct and may be attributed to the fact that the cell walls are negatively charged, and the SPIONs are positively charged. Figure 3b–d clearly shows that SPIONs near the nucleus are located inside the cell, while the SPIONs at the cell margins are more likely to be attached to the cell and therefore block immunofluorescence of the cytoskeleton. SPIONs can be internalized into the cells through various uptake mechanisms, depending on size, shape, and charge [40]. We assume that macropinocytosis is the uptake mechanism, as several studies indicate that aggregates of cationic nanoparticles with a hydrodynamic size similar to our SPIONs were taken up by macropinocytosis into tumor cells, while smaller aggregates are more likely to be taken up by clathrin-mediated endocytosis [15,41,42,43]. Calero et al. found that, while small aggregates were taken up within the first hour using the clathrin-dependent path and stayed near the cell membrane, larger SPION aggregates were taken up after 6–24 h via macropinocytosis and accumulated near the nucleus [42]. This accurately describes the process depicted in Figure 4a of cells taking up SPIONs mainly after 6–24 h and further confirms macropinocytosis as the most probable uptake mechanism. The fact that the SPIONs attach to or are taken up by the cells provides the prerequisite for the targeted delivery of drugs to or into the cells via modified SPION-APTES, such as previously carried out functionalizations [8,44].

The evolving uneven distribution of SPIONs in cells that appears in Figure 3a, as well as Figure 4a, is most likely attributed to three mechanisms. First, during cell division, the SPIONs distribute unevenly among the daughter cells (Figure 4b). Second, before reaching confluence, cells at the edges of the growing colony are exposed to more SPIONs than those inside the colony and therefore internalize more SPIONs. Third, cells containing many SPIONs appear to divide less frequently than their particle-less counterparts. We were able to make these observations mainly thanks to the novel approach of observing the interaction between cells and SPIONs using 24-well microscopy. The uneven distribution could be an artifact of cell culture, but it is also important to consider that in tissues, cell divisions and surfaces could cause uneven distribution of nanoparticles to cells.

It has already been studied that SPIONs can cause cytotoxicity through oxidative stress, as they can promote the formation of reactive oxygen species (ROS) [20,45,46]. This effect intensifies under ionizing radiation; thus, some SPIONs sensitize the cells to ionizing radiation [19,47]. We found that no toxicity occurred in most cell lines at 20 µg Fe/mL (Figure 5b–h). In Figure 5e–g, a slight divergence between the control group and the SPION group can be seen at 0 Gy, but this does not become significant. This is in agreement with the results of Akal et al., who found no toxicity of SPIONs coated with APTES at concentrations up to 200 µg Fe/mL [48]. Even after irradiation with up to 8 Gy, no significant difference can be seen in most cell lines. There are two exceptions: Cal33 cells are sensitive to SPIONs at 8 Gy and HSC-4 cells at 2 Gy. It is important to remember that in radiotherapy, very small effects can become relevant through fractioned treatment with high numbers of repetitions. However, contrary to the hypothesis of this study, APTES-SPIONs appear to be only weak radiosensitizers on their own. The decisive factor will therefore be which chemicals are bound to the SPIONs via APTES.

The cytostatic effect of the SPIONs varied greatly between the different cell lines: while the SPIONs had no effect at all on the growth rate of RPMI 2650 cells, the growth of Cal33 cells was significantly slowed down (Figure 6a,d). This effect was evident across all dose levels. Together with the results of the cytotoxicity assays, where Cal33 also showed the stronger effect, it gives the impression that the sensitivity to the SPION-APTES strongly depends on the cell line. This is supported by Akal et al., who also found that SPIONs coated with APTES of various concentrations slowed the growth of a tumor cell line but not that of the control cell line. The reduced cell division described by the slowed growth fits with the previously described observations that cells containing SPIONs divide less, contributing to uneven nanoparticle distribution.

Figure 6h suggests that the SPIONs exert a slight cytostatic effect on all cell lines, as it is unlikely that the doubling time of the control group is always the lowest by chance. However, this effect is not significant for all cell lines. Most of the cell lines showed a significant difference between the unirradiated control group and nanoparticle group. However, this difference was mostly rendered non-significant when both groups were irradiated with 2 Gy (Figure 6i). There is little overall literature on whether SPIONs slow cell growth, because most safety studies on SPIONs are limited to cytotoxicity, making it difficult to compare the observed effects.

In summary, the SPIONs themselves have no significant cytotoxic effect, but they can slightly enhance the cytotoxic effect of radiation on some cell lines. Despite the apparent low toxicity, the SPION-APTES appear to have a cytostatic effect on some cell lines, although it is worth noting that the effect varies greatly between cell lines.

SPION-APTES are not effective radiosensitizers per se. Nevertheless, we were able to make an important contribution to better illustrate and understand the behavior of cells towards the SPIONs and to identify the effects of the SPION-APTES on cells. The good compatibility of the SPION-APTES could also be advantageous in future studies or applications, as without SPION-APTES’s own effect on the cells, the effect of bound substances can be more easily researched and understood. This is an important basis for future studies with further functionalization of these SPIONs, which we are sure hold great potential.

## Figures and Tables

**Figure 1 nanomaterials-13-00330-f001:**
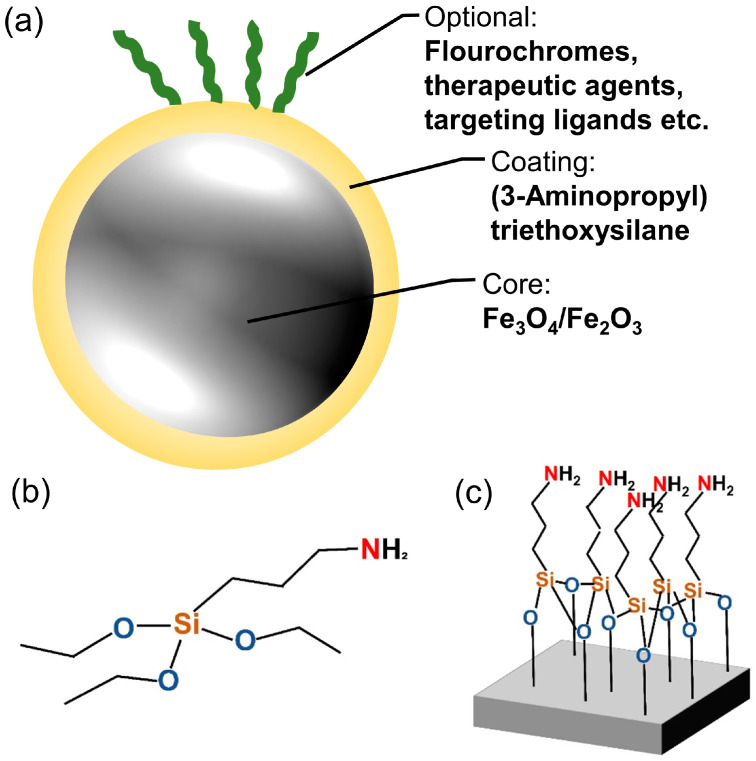
Superparamagnetic iron oxide nanoparticle (SPION) and coating structure. (**a**) Sketch of a SPION with (3-Aminopropyl)-Triethoxysilane (APTES) coating and optional additives. (**b**) Structural formula of APTES. (**c**) Structural formula of APTES on a surface.

**Figure 2 nanomaterials-13-00330-f002:**
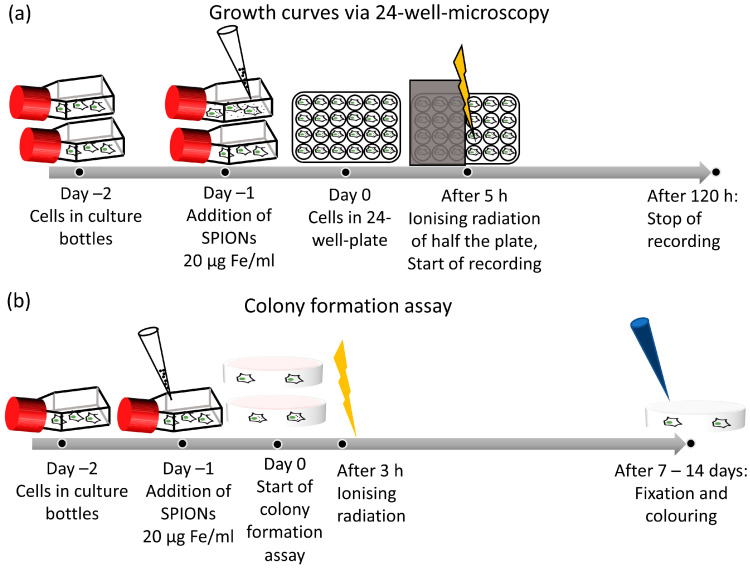
Experimental setup of growth analysis and colony formation assay. (**a**) Timeline of growth curve by 24-well microscopy. (**b**) Timeline of a colony formation assay.

**Figure 3 nanomaterials-13-00330-f003:**
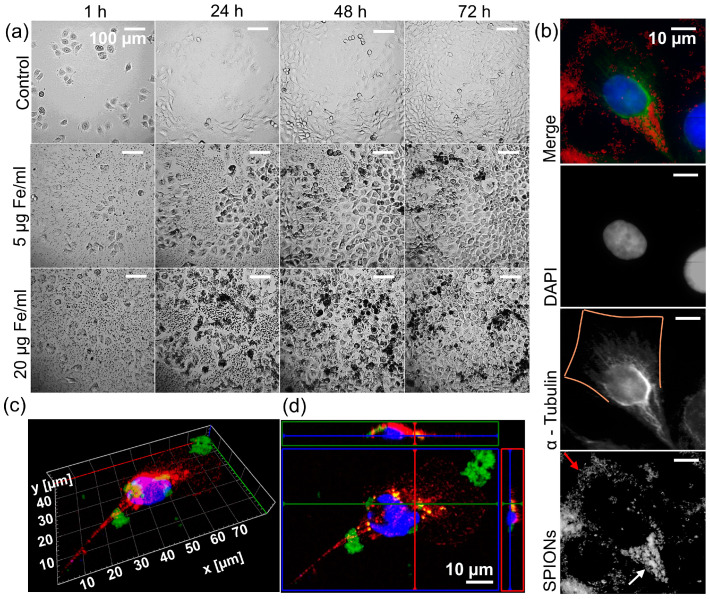
Intake of the SPIONs by the tumor cells. (**a**) Representative images of Cal33 cells, an HPV-negative (HPV-) head and neck squamous cell carcinoma (HNSCC) cell line, with different concentrations of nanoparticles (0, 5, and 20 µg Fe/mL) over time (1, 24, 48, and 72 h). For the full course of the interaction and depiction of 50 µg Fe/mL, see the video in Appendix A. (**b**) Immunofluorescence images of a cell from an HPV- HNSCC cell line RPMI 2650. Top to bottom: merge with DAPI (blue), α-Tubulin (green), nanoparticles (red); greyscale images of DAPI, α-Tubulin with orange line as estimated outline of the cell, nanoparticles captured with transmitted light microscope with red arrow on nanoparticles on cell margin, and white arrow on nanoparticles inside cell. (**c**) Three-dimensional immunofluorescence imaging of an RPMI 2650 cell with FITC-linked SPIONs (green), α-Tubulin (red), and DAPI (blue). (**d**) Orthogonal imaging of an RPMI 2650 cell with FITC-linked SPIONs (green), α-Tubulin (red), and DAPI (blue).

**Figure 4 nanomaterials-13-00330-f004:**
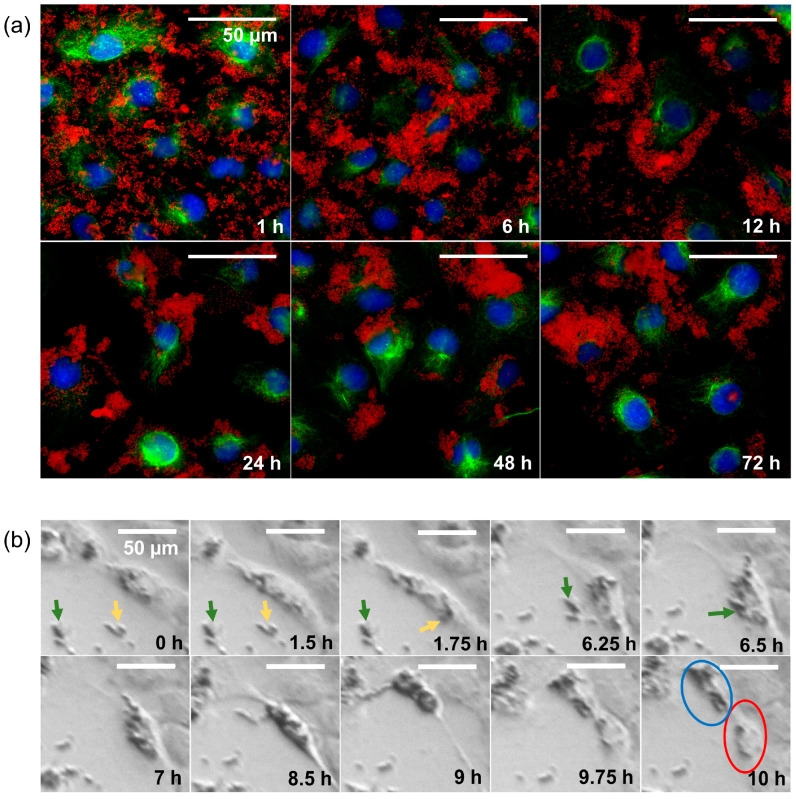
Behavior over time of HNSCC cells interacting with SPIONs. (**a**) Immunofluorescence imaging of RPMI 2650 cells with 20 µg Fe/mL over time (1, 6, 12, 24, 48, and 72 h); DAPI (blue), α-Tubulin (green), and nanoparticles (red, captured with transmitted light microscope). (**b**) Imaging of the cell division of an HSC-4 cell (HPV-HNSCC cell line) with SPION-APTES over the course of 10 h. Yellow and green arrows point to the aggregates of nanoparticles before uptake by the cell. The blue circle marks the daughter cell with a high content of nanoparticles, and the red circle marks the daughter cell with a low content of nanoparticles.

**Figure 5 nanomaterials-13-00330-f005:**
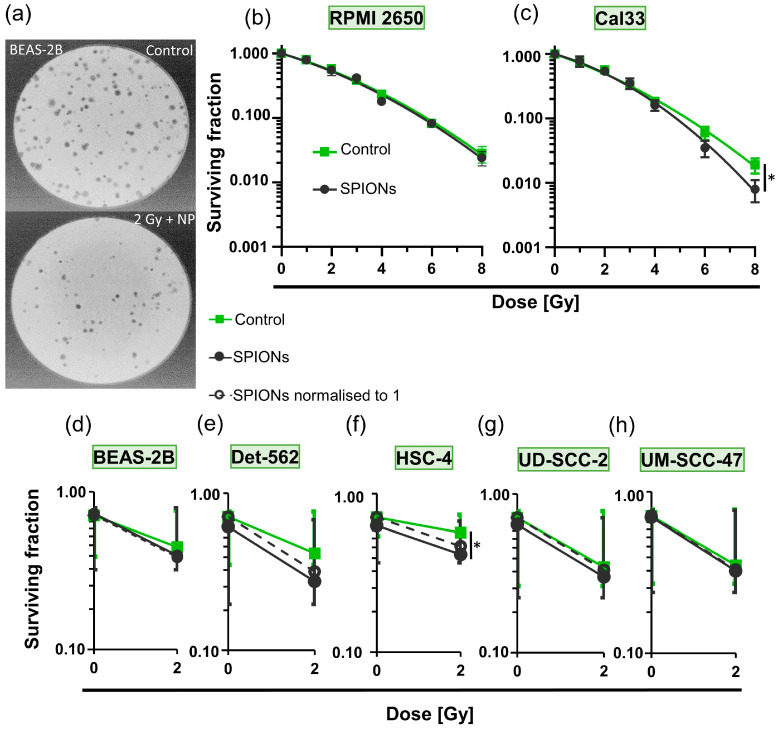
Detection of cytotoxic effects of SPION-APTES via colony formation assays. (**a**) Representative images of stained BEAS-2B (healthy bronchial epithelium cell line) colonies in Petri dishes during colony formation assay; top: control, bottom: 2 Gy of ionizing radiation and 20 µg Fe/mL of SPIONs. Logarithmic plots of survival fraction of (**b**) RPMI 2650 and (**c**) Cal33 cells; control group without SPIONs (green) and treated group with 20 µg Fe/mL SPIONs (black); ionizing radiation doses from 0 to 8 Gy; * describes significance with *p* = 0.05 between control group and nanoparticle group at 8 Gy, determined by Mann–Whitney U-test. Logarithmic plots of surviving fraction at 0 and 2 Gy without SPIONs (green) and with 20 µg Fe/mL of SPIONs (black); dashed lines represent nanoparticle groups normalized to 1; cell lines (**d**) BEAS-2B; (**e**) Detroit 562 (Det 562); (**f**) HSC-4; * describes significance with *p* = 0.05 between control group and nanoparticle group at 2 Gy, determined by Mann–Whitney U-test; (**g**) UD-SCC-2; (**h**) UM-SCC-47.

**Figure 6 nanomaterials-13-00330-f006:**
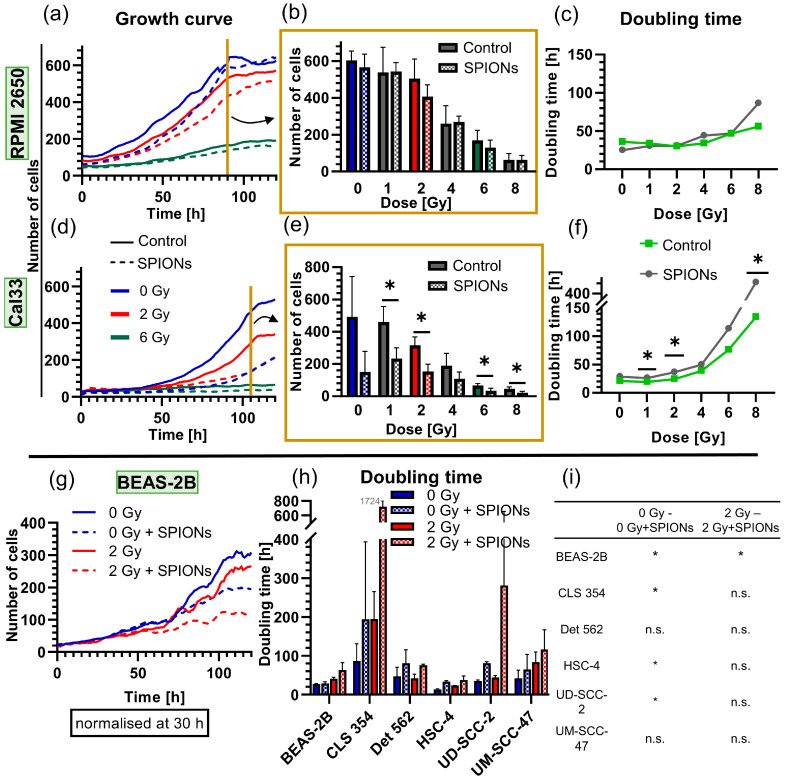
Analysis of the cytostatic effect of SPIONs at 20 µg Fe/mL via growth curves through 24-well microscopy. Growth curves of (**a**) RPMI 2650 and (**d**) Cal33 cells from 0 to 8 Gy, each with and without SPIONs over 120 h; for reasons of clarity, they are only depicted at 0, 2, and 6 Gy; dashed lines show nanoparticle group. Exemplary presentation of all doses of (**b**) RPMI 2650 cells at 90 h and (**e**) Cal33 cells at 105 h (end of exponential growth). Doubling times of the (**c**) RPMI 2650 and (**f**) Cal33 cells at doses from 0 to 8 Gy. (**g**) Exemplary growth curves of cell line (BEAS-2B) with and without 20 µg Fe/mL SPIONs and at 0 and 2 Gy over 120 h. (**h**) Doubling times of the cell lines BEAS-2B, CLS 354, Det 562, HSC-4, UD-SCC-2, and UM-SCC-47, each with and without SPIONs at doses of 0 and 2 Gy. (**i**) Significances between the different groups of (**h**); error bars indicate the standard deviation. * equals significance between control group and SPION group at respective radiation dose, determined by Mann–Whitney U-test with *p* = 0.05.

## Data Availability

Not applicable.

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
