# Peer review of "In Vitro Analysis of Superparamagnetic Iron Oxide Nanoparticles Coated with APTES as Possible Radiosensitizers for HNSCC Cells"

_nanomaterials, 2023, doi:10.3390/nano13020330_

Round 1
Reviewer 1 Report
Manuscript number: nanomaterials-2143889
Title: In vitro analysis of superparamagnetic iron nanoparticles coated with APTES as possible radiosensitizers for HNSCC cells
The present paper is interesting without any doubt. Nevertheless, minor additions should be added to the manuscript:
1) The Abstract part could be improved. It should contain some qualitative and quantitative results.
2) The introduction part can be further improved. The authors are suggested to include recent references in the introduction part. Previous studies on the applications of magnetic nanoparticles as radiosensitizers for HNSCC cells should be reported and discussed. As well, these studies could be used for comparison with the current results.
3) How the authors determined the optimal contents for SPIONs and APTES?
4) The characterization of samples before and after functionalization should be provided such as X-ray diffraction, TEM and particle size distribution, Fourier transform infrared spectroscopy (FTIR), and Magnetic properties (M-H and M-T).
5) The stability of prepared samples could be checked by Zeta size/potential, X-ray photoelectron spectroscopy (XPS), etc.
6) Error bars should be added for different results.
Reviewer 2 Report
In the manuscript by Emer et al, In vitro analysis of superparamagnetic iron nanoparticles coated with APTES as possible radiosensitizers for HNSCC cells. In this manuscript, authors have investigated the role of iron nanoparticles coated with APTES as possible radiosensitizers for HNSCC cells. The following comments and/or suggestions should be addressed before publication.
Format: Different fonts and font size have been used in the manuscript. So, please correct it all.
Introduction: Very short, need to elaborate more with previous studies along with various applications of SPIONs.
Authors need to mention about the applications of APTES and rational behind the selection of APTES as coating agent.
Materials methods: Divide them in subheadings, its very difficult to follow-up.
How did SPIONS were synthesized? How did coating was performed? How did you remove the uncoated SPIONs?
Results: Divide into subheadings.
As authors did not observe sensitizing effect from SPIONs, authors should have tried another approach by loading some chemotherapeutic agents into SPIONs and see the results.
Round 2
Reviewer 2 Report
Authors have addressed the majority of the comments and suggestions, recommended to accept it.